# Interactive Effect of Microplastics and Fungal Pathogen *Rhizoctonia solani* on Antioxidative Mechanism and Fluorescence Activity of Invasive Species *Solidago canadensis*

**DOI:** 10.3390/plants14131972

**Published:** 2025-06-27

**Authors:** Muhammad Anas, Irfan Ullah Khan, Rui-Ke Zhang, Shan-Shan Qi, Zhi-Cong Dai, Dao-Lin Du

**Affiliations:** 1School of Emergency Management, Jiangsu University, Zhenjiang 212013, China; anas.uaf@gmail.com; 2Institute of Environment and Ecology, School of the Environment and Safety Engineering, Jiangsu University, Zhenjiang 212013, China; ruizhang@163.com; 3Institute of Cotton Research, Chinese Academy of Agricultural Sciences, Anyang 455000, China; irfanullahkhan195@yahoo.com; 4School of Agricultural Engineering, Jiangsu University, Zhenjiang 212013, China; qishanshan1986120@163.com; 5Jiangsu Collaborative Innovation Center of Technology and Material of Water Treatment, Suzhou University of Science and Technology, Suzhou 215009, China; 6Jingjiang College, Jiangsu University, Zhenjiang 212013, China

**Keywords:** microplastics, invasive species, fungal pathogen, ecosystem, antioxidants, extracellular enzymes

## Abstract

Microplastics and invasive species, driven by anthropogenic activities, significantly disrupt ecosystems and microbial communities. This study investigated the interactive effects of biodegradable microplastics (polylactic acid, or PLA, and polyhydroxyalkanoates, or PHAs) and the fungal pathogen *Rhizoctonia solani* on the invasive plant *Solidago canadensis*. One plant of *Solidago canadensis*/pot was cultivated in forest soil amended with 1% (*w*/*w*) microplastics and/or *R. solani.* PLA exhibited greater toxicity than PHAs, reducing the plant height, root length, and biomass by 68%, 44%, and 70%, respectively. Microplastics impaired the maximum quantum yield of photosystem II more severely than *R. solani*. However, *S. canadensis* demonstrated adaptive antioxidative and extracellular enzymatic mechanisms under combined stresses. A heatmap analysis revealed a positive correlation between PHAs and plant growth traits, while a redundancy analysis explained the 15.96% and 4.19% variability for the first two components (r^2^ = 0.95). A structural equation model indicated the negative effects of morphology and physiology on biomass (β = −1.694 and β = −0.932; *p* < 0.001), countered by positive antioxidant contributions (β = 1.296; *p* < 0.001). These findings highlight complex interactions among microplastics, pathogens, and invasive species, offering insights into ecological management strategies under dual environmental pressures. Future studies should assess the long-term field effects and microbial mediation of these interactions.

## 1. Introduction

Terrestrial ecosystems, comprising a land-based community of organisms and their environment, are more vulnerable to anthropogenic activities than aquatic ecosystems. Among these impacts, the widespread distribution of microplastics by human activities has significantly disrupted microbial diversity and activity, altering the ecological balance of terrestrial ecosystems. Bouaicha et al. [1] reported that microplastics enhanced the pathogenicity of *Fusarium solani* against the root rot of tomato, while the plant-growth-promoting effect of *Trichoderma viride* was boosted. Microplastics served as source of carbon for microflora in the rhizosphere. Previous studies have showed that polyvinyl chloride, polycaprolactone, polystyrene–polyurethane, polyhydroxybutyrate, polyethylene terephthalate, and low- and high-density polyethylene enhanced the colonization of *Fusarium* spp. and *Trichoderma* spp. [2,3]. Similarly, the degree of toxicity of polyhydroxyalkanoates (PHAs) was negatively affected due to agglomeration. A high quantity of polylactic acid (PLA) microplastics in soil decreased the abundance of microorganisms and negatively affected the soil properties [4]. Baihetiyaer et al. [5] reported that PLA microplastics enhanced the oxidative stress and DNA damage, and changed the gene expression, of earthworms.

Plastics are widely used for their unique features like lightness, durability, ease of molding, portability, and elasticity for packaging, medical instruments, and mulch foil [6,7,8]. Approximately 300 million metric tons are produced annually; of that, 10% is recycled and 4725 million metric tons have been stocked since 2000 [8,9,10,11]. Moreover, it has been reported that 5–35 kg/ha of plastic has been used as mulch in agroecosystems [12]. It spreads through anthropogenic activities [13,14]. Conventional plastic was replaced with biodegradable plastics to mitigate the problem of degradation. However, their degradability is dependent on their composition, structure, and environmental factors like temperature, moisture, microbial activity, and soil properties [4,11]. However, they persist for centuries as micro- (<5 mm) and nanoplastics (<100 nm) [11,15]. PLA- and PHA-based plastics were reported to be biodegradable [16,17]. Microplastics are the emerging pollutant of the Anthropocene era and their contamination of ecological communities is increasing. They have been reported in biological processes, from cells to the ecosystem level [12,14]. Aquatic systems have been studied for microplastic contamination instead of terrestrial ecosystems, which were more vulnerable due to anthropogenic activities [18]. Microplastics are harmful for plants, animals, microorganisms, and humans [19,20,21,22,23]. They have been detected in the human body, especially in placentas, blood, breast milk, and other tissues [4,24,25]. Maize seedlings were tested for physiological, biochemical, and growth conditions using biodegradable and polyethylene microplastics and As-contaminated soil. Biodegradable microplastics significantly reduced the leaf area, chlorophyll content, and photosynthetic rate by 22%, 57%, and 78%, respectively [10].

Microplastics in soil disturb its chemistry, physiology, and microflora. They modify the soil aggregates and porosity, which are highly dependent on the shape, size, and quantity [12,13]. Microplastics increased the porosity of clay soil with a normal soil water content while increasing percolation and leaching loss in sandy soil [26,27]. Plant root growth was directly affected by microplastics via the clogging and mechanical damage of roots. Moreover, different types of microplastics influence the abundance of microflora in the rhizosphere. For example, PLA increases the abundance of *Ascomycota* and decreases that of *Mortierellomycota*, *Mucoromycota*, and *Basidiomycota* [1,4]. Microplastic PHAs were 83.9% less toxic to *M145 Streptomyces coelicolor* compared with PLA [28]. Similarly, the diversity of microplastics also harbors the root growth of invasive species more than that of native species [14].

Invasive species are also a result of the Anthropocene era, which has enhanced their growth on introduced sites. *Solidago canadensis* is a major invasive species in China and was introduced during the 1980s in Shanghai as an ornamental plant. It overcame native ecosystems by propagating through underground rhizomes [29]. It also exerts allelochemicals under the soil, efficiently capturing nutrients and distracting the microbial community [30,31]. Zhang et al. [32] reported that *S. canadensis* suppressed soilborne pathogens (*Pythium ultimum* and *R. solani*) through an allelopathic mechanism. The spore composition of arbuscular mycorrhizal fungi (AMF) and mycorrhizal colonization were negatively affected by exposure to the allelochemicals of *S. canadensis*. Moreover, a field abandoned by *S. canadensis* that was followed by tomato culture showed less root rot disease [32,33,34].

Terrestrial ecosystems have delicate and complex networks for their functions. Anthropogenic activities disrupt the balance of diversity distribution, niches, food, and shelter. Previous studies were performed to understand the impact of microplastics on the soil properties, fungal pathogens, and underground growth of invasive species. It has been reported that polyethylene microplastics were used as a substrate for *F. solani* proliferation in the rhizosphere of tomato seedlings. It enhanced, by 106%, the sporulation of *F. solani* and its pathogenicity [1]. Another study reported that the root growth of alfalfa decreased by 78% under *Rhizoctonia solani* inoculation. Conversely, microplastics showed higher catalase activity in leaves [35]. It is hypothesized that PLA is more problematic to plants than PHAs, and it may be enhanced by *R. solani* against invasive species. However, the combined interactive effect of biodegradable microplastics and *R. solani* on the typical invasive plant species *S. canadensis* has not been studied. In this experiment, we grew the invasive species *S. canadensis* under the dual pressure of biodegradable microplastics (PLA and PHAs) and *R. solani* to understand the defense mechanism and fluorescence response because the defense mechanism and fluorescence response are interconnected in plants under stress conditions. Although fluorescence directly reflects photosynthetic impairment, defense mechanisms mitigate secondary oxidative damage. Fluorescence impairment may be due to the translocation of microplastics, which may boost the accumulation of ROS in leaves and inhibit chlorophyll synthesis and Rubisco activity. Thus, a reduction in light use, dark respiration, and the prevention of electron transport between PSII and PSI may cease the growth of plant [36]. The objectives of this study were to assess (a) antioxidative and extracellular enzymatic activity of *S. canadensis*, both under and above the ground; (b) the impact of biodegradable microplastics and the soilborne pathogen *R. solani* on fluorescence; and (c) the synergistic or antagonistic effect of microplastics and a soilborne pathogen on the growth of *S. canadensis*. This study may provide insights to develop ecological management techniques, especially under the dual pressure of environmental pollution and pathogens, for invasive species and microplastic contamination by explaining the mechanisms that contribute to the invasive success of *S. canadensis* and, notably, the co-effects of the pathogenicity of fungi and environmental pollution.

## 2. Results

### 2.1. Morphological and Leaf Responses of Solidago Canadensis to Microplastics and a Pathogen

Microplastics significantly retarded the plant height and number of leaves of *S. canadensis* compared with the pathogen *R. solani*. Polylactic acid microplastics showed a greater negative effect than PHAs (Table 1). Plant height decreased by 67% under the PLAF treatment compared with the control. However, the number of leaves were statistically on a par with the control for *R. solani*. The leaf area was severely reduced under the combined application of PLAF with the lowest value (1.84 cm^2^). The specific leaf area indicated a reversal of the leaf area, and the maximum was observed for PLA. The specific leaf area describes how much of the leaf area is deployed to each unit of leaf biomass. Minimum leaf thickness was observed for PLA. The nitrogen content, relative chlorophyll, and flavanols were decreased by the stressors (Table 1).

### 2.2. Fluorescence and Leaf Greenness of Solidago Canadensis Under Biotic and Abiotic Stressors

The microplastic pollutants and fungal infection reduced leaf greenness and impaired the maximum quantum yield of photosystem II. The leaf images showed optimal fluorescence for the control (Figure 1), while a decreasing order was observed in the leaves of *S. canadensis* exposed to microplastic pollutants and the fungal pathogen (Figure 1). The highest SPAD value was observed for the control and decreased for the microplastics and fungal contamination (Figure 1a).

The maximum quantum yield of photosystem II was compromised under biotic and abiotic stressors. Minimum fluorescence (Fo) and maximum fluorescence (Fm) decreased under PLAF and PLA by 16% and 11%, respectively (Figure 1b,c). The maximum quantum yield of photosystem II declined for both types of microplastic and a similar decline was observed for the PLAF and PHAF treatments (Figure 1d). This revealed the physiological damage to photosynthetic apparatus by the microplastics and fungal pathogen.

### 2.3. Impact of Microplastics and a Pathogen on the Antioxidative and Extracellular Enzyme Activity of Solidago canadensis

The antioxidant response and extracellular enzymes were activated with the exposure to microplastics and the fungal pathogen *R. solani*. Catalase activity in the leaf was higher, particularly for PLA (Figure 2a). Similarly, it was higher in roots with the synergistic effect of *R. solani* and both microplastics (Figure 2c). Peroxidase activity upsurged in the leaf with exposure to the fungal pathogen (Figure 2b). However, its activity in root tissues was similar under PHA and PLA conditions and it was 67% and 71% higher than the control (Figure 2d). This indicated the strong response of *S. canadensis* to oxidative stress.

The extracellular enzymes β-1,4-N-acetyl-glucosaminnidase (NAG) and β-glucosidase (BG) were measured to check the activity of nitrogen and carbon cycles under the effect of microplastics and *R. solani* (Figure 2e,f). The highest activity for NAG was observed under PHAs compared with PLA. However, it was statistically at a par for PHAs without *R. solani* compared with PLA with *R. solani* (Figure 2e). *Rhizoctonia solani* increased the activity of BG under all microplastic conditions. The activity of BG was similar for both PHAs and PLA with and without *R. solani* (Figure 2f). Overall, microplastics and the fungal pathogen *R. solani* affected the activity of the extracellular enzymes NAG and BG.

### 2.4. Biomass and Root-to-Shoot Ratio of Solidago canadensis Under Both Stressors

The biomass and root length of the invasive species *S. canadensis* were negatively affected by microplastics (PHAs and PLA) and the fungal pathogen *R. solani*. Root length was strongly affected by PLA, along with/without *R. solani* (Table 2). It decreased by 28% and 21% under PHAs compared with the control (Table 2). Root biomass was more suppressed than shoot biomass. The shoot biomass of the invasive species *S. canadensis* was similar under PHA and PLA conditions without *R. solani*. However, the root biomass decreased (50%) with the combined effect of PLA and *R. solani* (Table 2). The highest root-to-shoot ratio was observed for PHAs, followed by PLA and *R. solani* (Table 2). *Rhizoctonia solani* decreased the total plant biomass by 39% compared with the control, while a 70% decrease was observed for the combined application of PLA and *R. solani* (Table 2). Significant differences in the mean value of all observations for each treatment are shown by different letters within the column at *p* < 0.05.

### 2.5. Relationship of Response Variables and Contribution to the Biomass of Solidago canadensis

Interactions between the various treatment variables for PHAs, PLA, and *R. solani*, and their subsequent impact on antioxidative and extracellular enzymatic activity, fluorescence, and growth of *S. canadensis*, are shown in Figure 3. The heatmap shows the correlation, along with the clustering of observed parameters and treatment variables. PHA microplastics with and without *R. solani* were positively associated with growth parameters such as the plant weight, shoot weight, root length, plant height, and leaf area (Figure 3a).

The redundancy analysis explained the variability for RDA1 (15.96%) and RDA2 (4.19%). Plant height and plant weight showed a positive relationship with the fluorescence parameters, while antioxidative enzymes and extracellular enzymes had a negative relationship with the observed parameters for shoot weight, fluorescence, and plant height (Figure 3b). The r^2^ value of 0.952 showed the fitness of the model and the influence of plant growth dynamics and biochemical responses.

Structural equation modeling revealed the direct and indirect relationships among the variables (Figure 4). *Rhizoctonia solani* and microplastic stressors affected changes in plant biomass, either directly or indirectly, by influencing morphology, physiology, enzyme, and chlorophyll fluorescence variables (Figure 4). Notably, a negative effect of morphology and physiology on plant biomass was observed (β = −1.694 and β = −0.932; *p* < 0.001), while antioxidants demonstrated a positive effect (β = 1.296; *p* < 0.001). These findings underscore how both biotic stressors (*R. solani*) and abiotic pollutants (microplastics) can contribute to a decrease in biomass, potentially undermining the growth and establishment of *S. canadensis* in affected terrestrial ecosystems.

## 3. Discussion

Microplastics and *R. solani* negatively affected the physiological, biochemical, and growth mechanisms of *S. canadensis*. The combination of both biotic and abiotic stressors have a potential implication to limit the growth of the invasive species *S. canadensis*.

Our results demonstrated that the maximum quantum yield of photosystem II was impaired by microplastics and *R. solani*. It was more pronounced under the combined application of PHAs/PLA with *R. solani* (Figure 1; Table 1). This suggests that microplastics change the chemical and physical properties of soil by agglomeration, choking pore spaces, nutrients, and microbial activity, whilst enhancing the pathogenicity of *R. solani* [4]. Leaf functions, like the leaf area and chlorophyll content, describe the activity of photosynthetic apparatus [10,37]. Specific leaf area describes the thickness of a leaf under stress conditions. It may reduce the nutrient uptake and photosynthetic rate, as evidenced by the decrease in nitrogen content, chlorophyll content, and the maximum quantum yield of photosystem II (Table 1 and Figure 1d). Our results are in line with Sunet al. [10], where the leaf area, chlorophyll content, and photosynthesis activity of maize plants decreased under the application of microplastics. Moreover, Shi et al. [38] reported that the SLA of invasive species was higher under the application of nutrients and microplastics.

The growth of a fungus largely depends on environmental changes such as nutrient availability, space, and disruption in binding mechanisms due to microplastics [39,40]. Furthermore, microplastics are a source of carbon for microorganisms [41,42]. The pathogenic impact of a fungus depends on the size of microplastics and spores [1]. Microplastics enhanced the pathogenicity of *F. solani.* This might be due to a large spore size, which provides greater adhesion to microplastics [1]. Polyhydroxyalkanoates (PHAs) degrade 7.4 times more compared with PLA, and PLA provides more adhesive sites for microorganisms [4]. In this study, a greater pathogenicity of *R. solani* was observed with PLA. *R. solani* extensively changed the anatomy and reactive oxygen species in chloroplasts and decreased photosynthetic efficiency in rice [43].

The antioxidative mechanism is activated to mitigate reactive oxygen species in response to external or internal stimuli. In this study, catalase and peroxidase activities increased with exposure to PHA and PLA microplastics and *R. solani* (Figure 2). This suggests that *S. canadensis* activated its antioxidative mechanism to scavenge the reactive oxygen species in roots and as well as leaves. Although antioxidative defense mechanism was activated due to microplastics and *R. solani* stress conditions. However, the magnitude of ROS production was exceeded and could lead to residual oxidative damage. This imbalance contributed to the observed reductions in plant height, root length, and biomass (Table 2), suggesting that the antioxidative response was only partially effective in mitigating stress. Previous studies have also reported that microplastics enhanced catalase, urease, and acid phosphatase activities by 149%, 234%, and 14% [44,45,46]. Biodegradable microplastics increased CAT activity more than polyethylene microplastics in maize—by 79%—when both were applied at the rate of 1% [10]. This study also depicted higher CAT activity for microplastics compared with the control. Oxidative damage to the lipid membrane is caused by an overproduction of H_2_O_2_, and it is reduced by POD into water and oxygen [47]. The production of POD increased with exposure to microplastics [10,48]. In this study, antioxidative activity was higher for microplastics compared with the control. However, it was higher for PHA microplastics. This might be due to the higher degradation of PHA compared with PLA [4].

Environmental changes influence the carbon and nitrogen cycles in soil. In this study, the activities of the extracellular enzymes NAG and BG were measured to detect the impact of microplastics and *R. solani* on nitrogen and carbon cycles in the rhizosphere of *S. canadensis*. The activity of NAG was related to nitrogen transformation, which was highly influenced by both microplastics. This suggests that microplastics, particularly PLA, may disrupt nitrogen cycling by impairing NAG-mediated chitin degradation, a key process in nitrogen release from organic matter. The lower nitrogen availability could directly contribute to the reduced leaf nitrogen content observed in *S. canadensis* under the PLA treatment. Previous studies also reported that the presence of microplastics significantly altered the nitrogen cycles due to changes in NAG activity [49,50]. On the other hand, the activity of BG was higher under biotic and abiotic stressors. This supports the hypothesis that *S. canadensis* had faster growth and deposited more litter, which may increase the activity of extracellular enzymes [31,34]. It suggests that the microbial activity increased with an increased substrate. However, the lack of a proportional increase in plant growth (e.g., biomass) suggests that the additional carbon might be diverted toward stress responses (e.g., antioxidant production) rather than growth.

The invasive alien species *S. canadensis* has strong growth capability due to the ‘novel weapon’ hypothesis [32]. It has a strong allelopathic nature and changes the microbial community of invaded ecosystems [29,31]. The fungal pathogen *R. solani* could not show a strong suppressive effect compared with the microplastics. Tomato seedlings exposed to an extract of *S. canadensis* as well as *R. solani* showed 31% damping off and mortality. However, when they were exposed to only *R. solani*, the damping off rate was 80% [32].

Above ground part of plant fix photo energy and supplies to belowground part. While roots absorb mainly water and nutrients and transport the above ground part of plant. The root structure describes the growth patterns of plants. Coarse and thick roots are less absorptive than thinner roots [14,51,52]. Microplastics affected the root growth of *S. canadensis* (Table 2). This suggests that the invasive plant species was unable to absorb enough nutrients under polluted soil as well as the extensive secretion of allelochemicals in a large soil volume. Plant height and biomass also decreased due to the insufficient uptake of nutrients by the roots. Overall, the growth of *S. canadensis* was negatively affected by different stressors in this study. Microplastic diversity and plant diversity showed different growth interactions [14,53].

The negative effects of microplastics and *R. solani* on plant growth were largely mediated by changes in morphology, physiology, and enzymatic activity. Interestingly, the positive effect of antioxidants on plant biomass suggests that, despite the detrimental impacts on growth, the antioxidative mechanism played a crucial role in mitigating oxidative stress (Figure 4). However, the overall negative impact on biomass underscores the overwhelming influence of the combined stressors on plant performance. Moreover, the synergism of both stressors might have been due to the dual mechanism of the microplastics. The microplastics served as source of carbon for *R. solani* and enhanced soil porosity due to agglomeration. This may have provided favorable conditions for hyphal growth and prolonged the persistence of the fungal pathogen. It resulted in a growth reduction in *S. canadensis*, which was in line with other research [4,9,35].

In this study, we focused on biodegradable microplastics; prior work suggests that non-biodegradable polyethylene may exert comparable stress on plants through physical damage to roots, soil intermission, and nutrient leaching [20,21]. However, the fast degradation of biodegradable PLA and PHAs could intensify short-term microbial shifts, whereas the persistence of polyethylene may lead to chronic soil alterations [18]. It was reported in a recent study that conventional polyethylene microplastics inhibited the mobility of arsenic in paddy soil, while biodegradable microplastics enhanced the mobility and methylation of arsenic in the same paddy soil [54].

These findings have important ecological implications. Microplastics changed the interaction between the fungal pathogen *R. solani* and plant roots. When the plant interacted with both biotic and abiotic stressors, the aggressive effect of PLA was higher compared with PHAs, which impaired the growth of *S. canadensis* in polluted environments. This might have been due to PLA degrading more slowly than PHAs.

This may, initially, seem beneficial to limit its invasiveness. However, ecological consequences warrant careful consideration. The suppressed growth of *S. canadensis* could alleviate competitive pressure on native species for a time, but the disruption to soil enzymatic activities (e.g., NAG and BG) and shifts in microflora may change nutrient cycling and soil health. These interactive changes between fungal pathogens and biodegradable microplastics may significantly impact the control of invasive species, upsetting the delicate balance of a terrestrial ecosystem and producing unidentified consequences for plant growth. Future studies should be long-term and field-based. They should evaluate whether these stressors selectively disadvantage invasive species over native species or inadvertently exacerbate ecosystem degradation.

## 4. Materials and Methods

The seeds of *S. canadensis* were collected from the riverside of the Yangtze River, Zhenjiang, China. The collected seeds were stored prior to being raised in a nursery at 4 °C [55]. The seeds were sown in wet sand and kept wet to allow germination and to raise the nursery. The seedlings were ready to transplant at the age of 30 days [29].

The microplastics used in the experiment consisted of biodegradable and pristine PLA and PHA microplastics (Zhonglian Plastic, Taizhou, China). PLA is more toxic than other biodegradable microplastics, e.g., PHAs [4]. The properties of the microplastics were determined as per our previous study, and the microplastic particles were spherical with a rough surface [56]. Further uniformity of the microplastics was ensured by sieving and cleaning to remove aggregates or fine particles. The size of particles ranged from 150 to 185 μm, commonly observed in microplastic-contaminated soils and selected based on prior studies. The particles of microplastics were cleaned with methanol and air-dried for the complete evaporation of methanol [9].

We used forest land soil from the Jiaoshan mountain area located at bank of the Yangtze River, Zhenjiang, China. The soil properties were measured according to Kama et al. [57] and soil pH was 6.55 ± 0.05, with organic matter of 1.95%, total nutrients of 1.8%, a water content of 23%, and electrical conductivity of 1.69 ds/m. This was because one-third of global protected land is under intense human pressure [58]. The culture media were prepared according to our previous study [35]. Briefly, soil was collected from the riverside and sieved through a 2 mm sieve to remove debris and stones. This was air-dried, and pots were filled with 400 g of forest land soil sieved with a 2 mm sieve. For the microplastic (M) treatment, the pots were amended with 1% (*w*/*w*) biodegradable microplastics, either polylactic acid (PLA) or polyhydroxyalkanoates (PHAs), and mixed well [52,57]. The filled pots were kept for two weeks to set down the soil. The concentration of microplastics was consistent with environmentally relevant estimates under different levels of contamination [12,59,60,61,62].

*Rhizoctonia solani* was isolated from diseased alfalfa plants and roots with typical symptoms and were air-dried. Small segments of roots were dipped in ethanol (70%) for 5 s followed by a mercuric chloride solution (0.1%) for 30 to 60 s. After thoroughly washing with deionized water, the samples were surface-dried under a vacuum hood and placed on PDA plates. The cultural and hyphal morphological characterization of *R. solani* was performed on the basis of septate hyphae, multinucleate cells, right-angled hyphal branching, a lack of conidia, and the production of monoploid cells [63].

A millet-seed-based inoculum of *R. solani* was prepared as described by Fang et al. [64]. Millet (*Panicum miliaceum*) seeds were soaked in deionized water for 10 h and autoclaved after rinsing at 121 °C for 20 min. Five-day-old colonies of *R. solani* were sub-cultured on autoclaved millet seeds and incubated at 22 °C in darkness for 2 weeks and mixed well for uniform colonization at an interval of 2 days.

This study was conducted at the Institute of Environment and Ecology, School of Environment and Safety Engineering, Jiangsu University, Zhenjiang, China. The experiment was conducted under controlled conditions at 22–26 °C, 80% humidity, and 14 h light and 10 h dark photoperiods. One seedling of *S. canadensis* was transplanted into each pot (Figure 5). The millet-seed-based inoculum was used according to Akber et al. [65], with 5 g inoculum mixed into 1000 g soil. Inoculated seeds of millet (0.5% *w*/*w*) were used for the *R. solani* fungal treatment after transplanting the seedlings. There were six treatments, each with eight replicates, including a control (without microplastics and a fungus), a fungal inoculation of *R. solani*, polyhydroxyalkanoate microplastic (PHA), PHAs combined with a fungus (PHAF), polylactic acid microplastic (PLA), and PLA combined with a fungus (PLAF).

Plant height was recorded by measuring from the base of the plant to the first fully opened leaf with a measuring tape, and number of leaves per plant were counted. Plants were harvested and their roots were washed. The harvested fresh roots were scanned and WinRHIZO software (regular version)was employed to measure the root length. For the dry biomass, plants were harvested and separated into above-ground and below-ground parts. These separated parts were kept at 70 °C up to a constant weight [66].

Leaf greenness, leaf nitrogen content, relative chlorophyll, and flavanols were measured by a SPAD meter and a Multi-Pigment-Meter (MPM-100). The leaf area (LA) was measured by employing the YMJ-CH intelligent leaf-area system (Topu Yunnong Technology, Zhejiang, China). Briefly, fully expanded leaves were excised from plants and their images were captured by the YMJ-CH intelligent leaf-area system. The corresponding readings were recorded. These leaves were oven-dried to quantify their dry weight, and the specific leaf area (SLA) was calculated using the following formula:SLA = LA/dry weight(1)

A FluorCam system was used to measure the maximum quantum yield of photosystem II (Fv/Fm). The open FluorCam system (Photo Subsystem Instrument; Drásov, Czech Republic) was used according to manufacturer’s protocol. The FluorCam generated 720 × 560 pixel images with a 12 bit resolution at 20 images/sec. It captured images of detached leaves. After imaging, FluorCam7 software was used to analyze the images and obtain the value of Fo (initial fluorescence) and Fm (maximum fluorescence) [67]. The maximum quantum yield of photosystem II (Fv/Fm) was measured by the following formula:Fv/Fm = (Fm − Fo)/Fm(2)

Plants activate their antioxidative mechanism in response to oxidative stress due to external stimuli such as microplastics [56]. Antioxidative enzymatic activity was determined by grinding 0.2 g of fresh tissue samples in an ice-cold phosphate buffer (pH = 7.0). A supernatant was obtained by removing debris after homogenizing at 12,000 rpm for 10 min at 4 °C. It was further used to determine the enzyme activity [68]. The reaction mixture for peroxidase (POD) activity was prepared as 0.1 mL supernatant (enzyme extract) and 1 mL 0.3% H_2_O_2_ and 0.2% guaiacol. The activity was observed at a 470 nm wavelength using a spectrophotometer. Similarly, the reaction mixture for catalase (CAT) activity comprised 1 mL 0.3% H_2_O_2_, 1.9 mL H_2_O, and 0.1 mL enzyme extract. The spectrophotometer wavelength was 240 nm [69,70].

Rhizosphere soil was collected to measure the activity of the extracellular enzymes NAG and BG. An ELISA kit was used to measure the activity of NAG. The initial reaction mixture contained a 50 μL standard, 40 μL sample dilution, and 10 μL diluted sample. An HRP conjugate reagent was added at 100 μL, except for a blank, and incubated for 60 min at 37 °C. After washing with a buffer, chromogen solutions A and B were added at the rate of 50 μL and covered for 15 min at 37 °C. Finally, a stop solution (50 μL) was added, which turned the blue color to yellow, and absorbance was read at 450 nm. The activity was measured by calculating the regression line between the standard and absorbance value, then multiplying this with the dilution factor. The activity of BG was measured according to Sainju et al. [71].

An analysis of variance (ANOVA) was performed with a factorial design, and a Tukey HSD test was used to differentiate the mean values. Both tests were performed using Statistix 8.1 software. Data were transformed to a normal distribution by using z-score normalization for the heatmap cluster analysis, redundancy analysis, and structural model equation [72]. The heatmap and redundancy plots were created by using the online functions of ChiPlot. However, Origin Lab 2018 software was used for the scatter plot.

## 5. Conclusions

Biodegradable microplastics (PLA and PHAs) and the fungal pathogen *R. solani* synergistically impaired the growth, physiology, and biochemical responses of the invasive species S. canadensis. PLA significantly reduced the plant height, root elongation, and biomass and depicted a stronger inhibition than PHAs. The disruption to photosystem II efficiency and alterations in enzymatic activity highlighted the complex stress response of the invasive species s. canadensis under biotic and abiotic stressors. Despite these challenges, invasive species showed an adaptive nature by activating antioxidative and extracellular enzymatic mechanisms. This may also have caused the flexibility of S. canadensis to co-occurrence with stressors. This study enhances the understanding of plant stress physiology, and provides evidence of the ecological risks posed by microplastic pollution and invasive species. Further long-term investigations into plant-pathogen–microplastic interactions might provide a broader implication for biodiversity and ecosystem stability. This is crucial for developing effective management strategies to mitigate the ecological consequences of microplastic pools and the invasion process.

## Figures and Tables

**Figure 1 plants-14-01972-f001:**
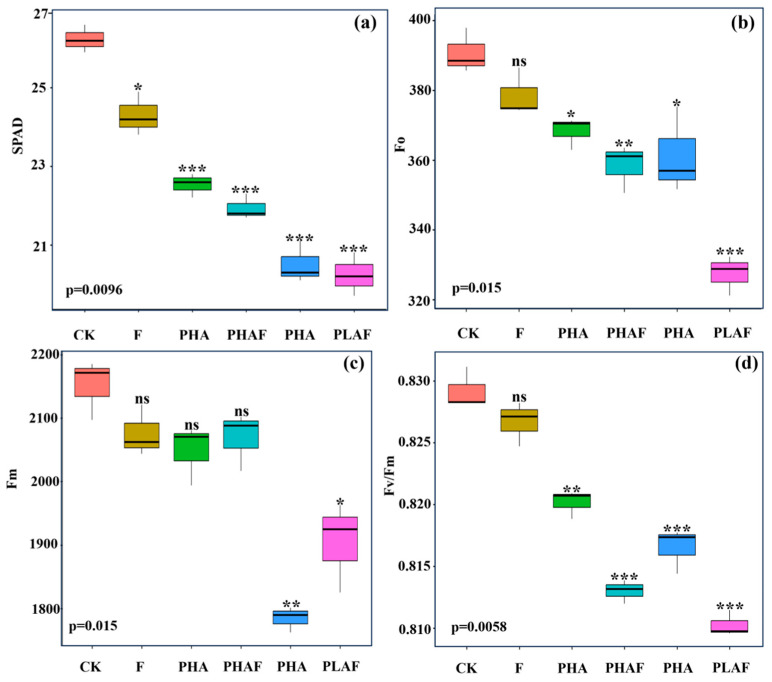
Leaf greenness and fluorescence indices of *Solidago canadensis* for different microplastic and pathogen conditions. Box plot shows the mean value of leaf greenness (**a**), initial fluorescence (**b**), maximum fluorescence (**c**), and maximum quantum efficiency of photosystem II (**d**). CK: control without *Rhizoctonia solani* and microplastics; F: fungal pathogen *Rhizoctonia solani*; PHA: polyhydroxyalkanoate microplastic; PHAF: polyhydroxyalkanoate microplastic and *Rhizoctonia solani*; PLA: polylactic acid microplastic; PLAF: polylactic acid microplastic and *Rhizoctonia solani*. Asterisks represent the statistical difference or similarities between the control and treatments e.g., ns: non-significant at *p* < 0.05; *: significantly different at *p* ≥ 0.5; **: significantly different at *p* ≥ 0.01; ***: significantly different at *p* < 0.01.

**Figure 2 plants-14-01972-f002:**
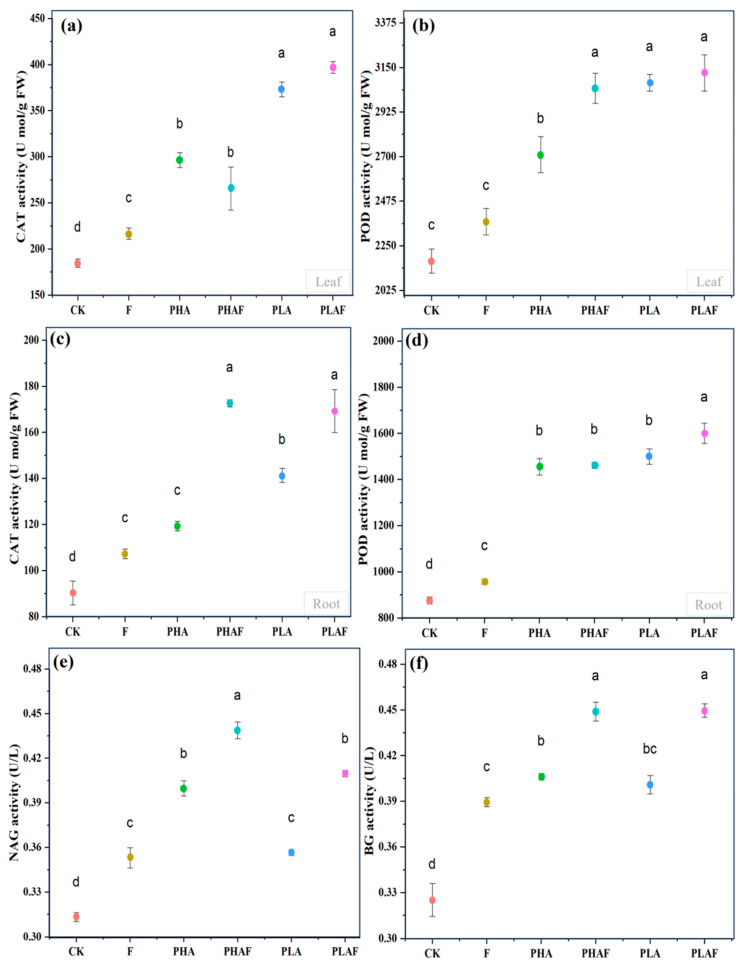
Antioxidant and extracellular enzyme activity in the leaf, roots, and rhizosphere soil of *Solidago canadensis*. (**a**) Catalase activity in the leaf, (**b**) peroxidase activity in the leaf, (**c**) catalase activity in roots, (**d**) peroxidase activity in roots, (**e**) β-1,4-N-acetyl-glucosaminnidase (NAG) activity in rhizosphere soil, and (**f**) β-glucosidase (BG) activity in rhizosphere soil. Different letters represent the statistical differences of response variables among biotic (*Rhizoctonia solani*) and abiotic (microplastics) stressors.

**Figure 3 plants-14-01972-f003:**
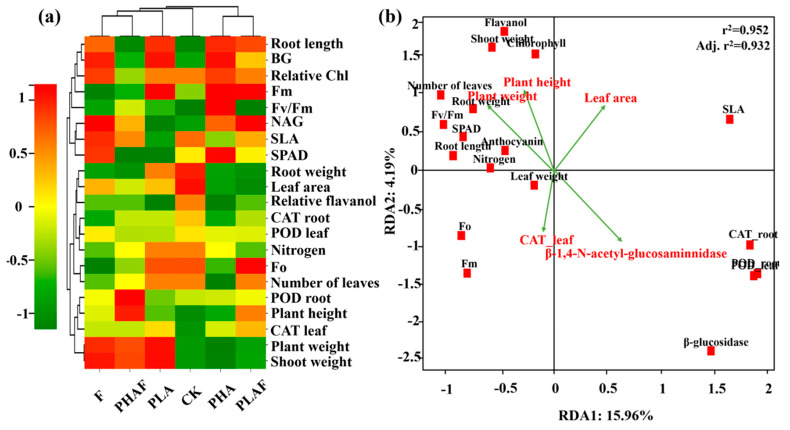
Heatmap representing the grouping of treatment and response variables. Red color shows a positive relationship and green color shows a negative relationship. The dendrogram on the upper and right side shows the clustering of treatment and response variables, respectively (**a**). The redundancy analysis shows the relationship between response variables. Vectors represent the explanatory variables and red squares show the plant variables (**b**).

**Figure 4 plants-14-01972-f004:**
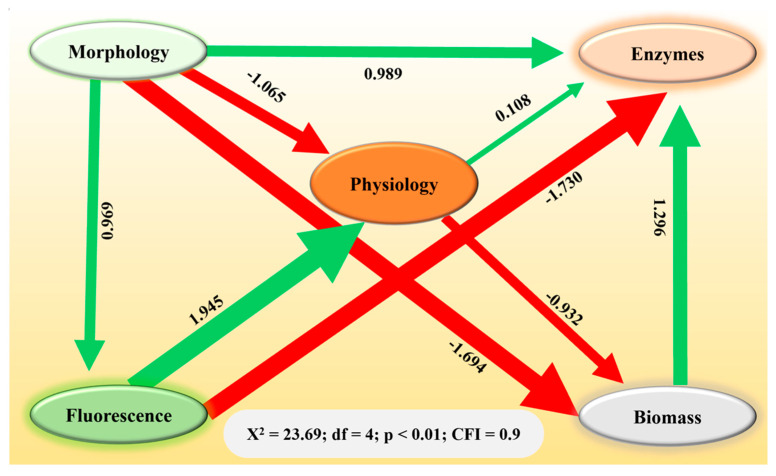
Structural equation model representing the contribution (direct and indirect) of morphology, physiology, fluorescence, and enzymes to the biomass of *Solidago canadensis*. Heads of arrows show the direction of contribution. Green and red colored arrows represent positive and negative contributions, respectively.

**Figure 5 plants-14-01972-f005:**
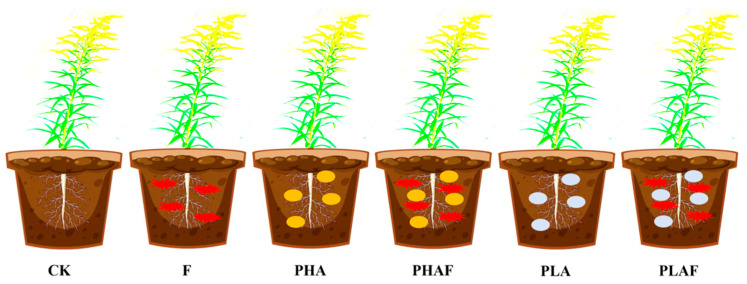
Schematic methodology designed for conducting experiments. CK: control without *Rhizoctonia solani* and microplastics; F: fungal pathogen *Rhizoctonia solani*; PHA: polyhydroxyalkanoate microplastic; PHAF: polyhydroxyalkanoate microplastic and *Rhizoctonia solani*; PLA: polylactic acid microplastic; PLAF: polylactic acid microplastic and *Rhizoctonia solani*.

**Table 1 plants-14-01972-t001:** Morphology and leaf physiology of invasive species (*Solidago canadensis*) under different microplastic and pathogenic conditions.

Treatment	Plant Height (cm)	Number of Leaves	Leaf Area (cm^2^)	SLA (cm^2^/g)	Nitrogen (mg/g)	Relative Chlorophyll	Relative Flavanols
CK	18.86 ± 0.38 a	30 ± 0.55 a	3.98 ± 0.03 a	0.36 ± 0.01 d	2.53 ± 0.27 a	0.33 ± 0.02 a	0.073 ± 0.01 a
F	18.17 ± 0.21 b	29 ± 0.45 a	3.77 ± 0.10 a	0.42 ± 0.01 cd	2.37 ± 0.05 ab	0.28 ± 0.01 ab	0.036 ± 0.00 b
PHA	8.53 ± 0.14 c	15 ± 0.52 b	3.20 ± 0.10 b	0.54 ± 0.03 b	2.3 ± 0.06 bc	0.26 ± 0.01 b	0.033 ± 0.01 b
PHAF	7.83 ± 0.06 d	13 ± 0.63 c	3.03 ± 0.19 b	0.45 ± 0.01 c	2.3 ± 0.09 bc	0.27 ± 0.02 b	0.027 ± 0.01 b
PLA	6.30 ± 0.08 e	13 ± 0.52 c	2.34 ± 0.18 c	0.65 ± 0.03 a	2.13 ± 0.05 cd	0.27 ± 0.01 b	0.027 ± 0.01 b
PLAF	6.13 ± 0.05 e	12 ± 0.52 c	1.84 ± 0.15 c	0.55 ± 0.01 b	2.07 ± 0.05 d	0.24 ± 0.01 b	0.23 ± 0.01 b

**Note:** CK: control without microplastics and *Rhizoctonia solani*; F: *Rhizoctonia solani*; PHA: polyhydroxyalkanoate microplastic; PHAF: polyhydroxyalkanoate microplastic and *Rhizoctonia solani*; PLA: polylactic acid microplastic; PLAF: polylactic acid microplastic and *Rhizoctonia solani*. Different letters and standard deviations within columns show the mean differences among treatments at *p* < 0.05.

**Table 2 plants-14-01972-t002:** Response of root length and biomass of invasive species (*Solidago canadensis*) to microplastics and a pathogen (*Rhizoctonia solani*).

Treatment	Root Length (m)	Shoot Weight (g)	Root Weight (g)	Root/Shoot	Plant Weight (g)
CK	2.82 ± 0.03 a	0.39 ± 0.01 a	0.04 ± 0.0 a	0.10 ± 0.0 c	0.43 ± 0.01 a
F	2.42 ± 0.07 b	0.23 ± 0.01 b	0.03 ± 0.0 b	0.14 ± 0.0 bc	0.26 ± 0.01 b
PHA	2.22 ± 0.06 c	0.14 ± 0.02 cd	0.03 ± 0.0 bc	0.21 ± 0.02 a	0.17 ± 0.02 cd
PHAF	2.04 ± 0.04 c	0.11 ± 0.01 d	0.02 ± 0.01 d	0.21 ± 0.04 a	0.14 ± 0.01 d
PLA	1.79 ± 0.06 d	0.19 ± 0.02 bc	0.03 ± 0.0 c	0.15 ± 0.02 bc	0.22 ± 0.02 c
PLAF	1.57 ± 0.03 e	0.12 ± 0.01 d	0.02 ± 0.0 e	0.16 ± 0.01 b	0.13 ± 0.01 d

**Note:** CK: control without microplastics and *Rhizoctonia solani*; F: *Rhizoctonia solani*; PHA: polyhydroxyalkanoate microplastic; PHAF: polyhydroxyalkanoate microplastic and *Rhizoctonia solani*; PLA: polylactic acid microplastic; PLAF: polylactic acid microplastic and *Rhizoctonia solani*. Different letters and standard deviation within columns show the mean differences among treatments at *p* < 0.05.

## Data Availability

The data will be made available upon special request to the corresponding author.

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
