# Peer review of "Interactive Effect of Microplastics and Fungal Pathogen Rhizoctonia solani on Antioxidative Mechanism and Fluorescence Activity of Invasive Species Solidago canadensis"

_plants, 2025, doi:10.3390/plants14131972_

Round 1
Reviewer 1 Report
Comments and Suggestions for Authors
This manuscript would benefit from a final thorough language edit. Carefully revise the entire text for spelling and grammatical errors. Pay special attention to the consistent and correct use of scientific notation and special symbols, such as the degree Celsius (°C) symbol.
Author Response
Respected Reviewer, Please see the attachment.

Reviewer 2 Report
Comments and Suggestions for Authors
Comments - Manuscript ID: plants-3676980
The manuscript investigates the combined effects of biodegradable microplastics (PLA and PHA) and the fungal pathogen Rhizoctonia solani on the invasive plant Solidago canadensis. The study is timely, given the rising concerns about microplastic pollution and invasive species management. The experimental design is robust, and the results are presented clearly. However, several areas require clarification, methodological refinement, and deeper discussion to strengthen the manuscript's impact and reproducibility.
- Lack of Clear Hypotheses: The introduction does not explicitly state the hypotheses being tested. A clear hypothesis (e.g., " PLA would exhibit stronger toxicity than PHA and synergize with R. solani to impair S. canadensis growth") would strengthen the study's focus.
- It is also difficult to understand why both PHA and PLA would persist in microplastics form for prolonged periods. Both of these are naturally synthesized compounds (polylactic acid and hydroxybutyric acid) and therefore, the prior evidence shows that they undergo rapid degradation in soil. This brings into question, the whole purpose of this study. Please justify.
- Overlap with Background: The background on microplastics and invasive species is thorough but could better integrate prior studies on their combined effects. For instance, are there any existing reports on how microplastics alter plant-pathogen interactions?
- Microplastic Characterization:
- The manuscript lacks critical details about the microplastics used (e.g., particle size distribution beyond "150–185 µm," surface properties, degradation state). Were they pristine or weathered? This significantly impacts toxicity.
- No information is provided on potential additives or contaminants in the microplastics, which could confound results.
- Soil Properties:
- Baseline soil characteristics (pH, organic matter, nutrient content) are omitted. These factors modulate microplastic and pathogen effects and are essential for reproducibility.
- Pathogen Inoculum:
- The fungal inoculum density (e.g., CFU/g soil) is not quantified, making it difficult to assess the consistency of pathogen pressure across replicates.
- Control Treatments:
- Were there controls for potential contaminants in microplastics (e.g., solvents used for cleaning)?
- Physiological Metrics:
- The increase in specific leaf area (SLA) under PLA (Table 1) is noted but not explained. Is this due to stress-induced leaf thinning? A brief discussion is needed.
- Antioxidant Activity:
- While catalase/peroxidase activity increased (Fig. 2), the manuscript does not clarify whether this response was sufficient to mitigate oxidative damage, given the significant biomass reduction.
- Extracellular Enzymes:
- The link between NAG/BG activity shifts and nutrient cycling (e.g., reduced NAG correlating with lower leaf nitrogen) is not explored. This is a missed opportunity to connect enzymatic changes to functional outcomes.
- Mechanistic Depth:
- The discussion does not address how microplastics might directly interact with R. solani. Do they serve as carbon sources? Alter hyphal growth? This is critical for understanding synergism.
- Comparisons with non-biodegradable microplastics (e.g., polyethylene) are absent, limiting the broader relevance of the findings.
- Ecological Implications:
- The manuscript frames reduced S. canadensis growth as negative, but could this actually benefit ecosystems by curbing invasion? Alternatively, could disrupted soil functions outweigh this benefit? These trade-offs are not discussed.
- Lab conditions may not replicate field dynamics (e.g., microbial community complexity, climate variability). This should be acknowledged.

Author Response

(The authors gave the same response as above.)

Reviewer 3 Report
Comments and Suggestions for Authors
Please read the attached file

Author Response

(The authors gave the same response as above.)

Round 2
Reviewer 2 Report
Comments and Suggestions for Authors
The authors have adequately addressed the earlier review comments. This manuuscript can be accepted for publication.